# Complete Genomic Characterization and Identification of *Saccharomycopsis*
*phalluae* sp. nov., a Novel Pathogen Causes Yellow Rot Disease on *Phallus rubrovolvatus*

**DOI:** 10.3390/jof7090707

**Published:** 2021-08-28

**Authors:** Xiaoxiao Yuan, Keqin Peng, Changtian Li, Zhibo Zhao, Xiangyu Zeng, Fenghua Tian, Yu Li

**Affiliations:** 1Department of Plant Pathology, College of Agriculture, Guizhou University, Guiyang 550025, China; xiaoxiaoyuanfungi@aliyun.com (X.Y.); pengkeqin@aliyun.com (K.P.); zbzhao@gzu.edu.cn (Z.Z.); xyzeng3@gzu.edu.cn (X.Z.); 2Engineering Research Center of Chinese Ministry of Education for Edible and Medicinal Fungi, Jilin Agricultural University, Changchun 130118, China; lct@jlau.edu.cn (C.L.); liyu@jlau.edu.cn (Y.L.); 3Institute of Edible Fungi, Guizhou University, Guiyang 550025, China

**Keywords:** edible and medicinal mushroom, de novo sequencing and assembly, Koch’s postulates, *Phallus rubrovolvatus*, *Saccharomycopsis* sp. nov

## Abstract

“Hongtuozhusun” (*Phallus rubrovolvatus*) is an important edible and medicinal mushroom endemic to Southwest China. However, yellow rot disease is a severe disease of *P*. *rubrovolvatus* that occurs extensively in Guizhou Province. It has caused major economic losses and hinders the development of the *P*. *rubrovolvatus* industry. In this study, 28 microorganism strains were isolated from diseased fruiting bodies of *P. rubrovolvatus* at various stages, two of which were confirmed to be pathogenic based on Koch’s postulates. These two strains are introduced herein as *Saccharomycopsis*
*phalluae* sp. nov. based on morphological, physiological, and molecular analysis. We reported a high-quality de novo sequencing and assembly of the *S*. *phalluae* genome using single-molecule real-time sequencing technology. The whole genome was approximately 14.148 Mb with a G+C content of 43.55%. Genome assembly generated 8 contigs with an N50 length of 1,822,654 bp. The genome comprised 5966 annotated protein-coding genes. This is the first report of mushroom disease caused by *Saccharomycopsis* species. We expect that the information on genome properties, particularly in pathogenicity-related genes, assist in developing effective control measures in order to prevent severe losses and make amendments in management strategies.

## 1. Introduction

*Phallus rubrovolvatus* (M. Zang, D.G. Ji and X.X. Liu) Kreisel, Phallaceae, is a basidiomycete endemic to the temperate regions of Southwest China [1,2]. It is known as “Hongtuozhusun”, “Flower of Fungi”, or “Wild Foods” in Chinese (Figure 1A). Its fruiting body is white, delicate, refreshing, with high nutritional, medical and economic value [3,4]. *Phallus rubrovolvatus* cultivation in Guizhou Province, certified as a geographical indication product of China, is an industry that has a lot of investment and is highly efficient. Multiple planting modes, including simple greenhouse planting, intelligent layer planting, and undergrowth planting, were established in Guizhou Province. According to the Department of Agriculture and Rural Affairs of Guizhou Province, the scale of *P. rubrovolvatus* cultivation in 2020 was estimated to have exceeded 80 million sticks.

Earthing is an essential process during the cultivation of *P**. rubrovolvatus*. It takes 4–10 months from earthing to harvest. During the process, the mushroom is vulnerable to many environmental microorganisms [5,6,7]. Diseases are common in *P**. rubrovolvatus* but it is difficult to detect them in the early stages. Among them, yellow rot disease has caused many epidemics over the past 20 years, which has resulted in major yield reductions and economic losses (Figure 1B–D). However, the cause of the disease has remained ambiguous [8].

*Saccharomycopsis* was introduced by Schiønning [9] as a member of the family Saccharomycopsidaceae [10]. *Saccharomycopsis* species are characterized by multi-polar budding and septate hyphae. Significant variations in the ascospore shape can lead to the false assignment of these species to other genera such as *Endomyces* and *Arthroascus* [11]. There are 19 species of *Saccharomycopsis* in Index Fungorum as (http://www.indexfungorum.org/Names/Names.asp, accessed on 26 August 2021). *Saccharomycopsis fibuligera* and *Saccharomycopsis cerevisiae* were developed as sourdough bread starters [12]. Additionally, *Saccharomycopsis*
*fibuligera* was reported as a specific biocontrol agent of ochratoxic molds (*Aspergillus ochraceus* and *Penicillium nordicum*) [13]. However, *Saccharomycopsis* has not been reported as a plant or mushroom pathogen. 

The present study reports for the first time on the association of the genus *Saccharomycopsis* with yellow rot symptoms on *P. rubrovolvatus* in China. The objective of the study was to identify the causal agent of yellow rot disease on *P. rubrovolvatus* in Guizhou Province, China. Herein, they are introduced as a novel species in the genus *Saccharomycopsis* (family Saccharomycopsidaceae) based on morphological and physiological evaluation and phylogenetic analyses. The genome of the causal agent was sequenced and annotated. We expect our findings to provide a reference for effective prevention and control of yellow rot disease on *P. rubrovolvatus*.

## 2. Materials and Methods

### 2.1. Pathogen Isolation and Maintenance

Diseased fruiting bodies of *P. rubrovolvatus* at three different degrees of disease (two samples/degree) were collected from a mushroom cultivation base, Nayong county (105°20′31″ N, 26°53′41″ E), Guizhou Province, China, in March 2020.

Each basidiocarp was first externally washed with running tap water. Next, the site of the disease was cut off with a cleansing knife and surface sterilized with 95% ethanol for 1 min, rinsed with sterilized distilled water twice, immersed in 75% ethanol for 30 s, and then rinsed in distilled water three times. The diseased tissue was crushed, immersed in sterilized water, and then subjected to gradient dilution. Low-titer spore suspensions were spread on a potato dextrose agar plate. For each sample, the single colonies formed by germination were re-isolated and purified after 4 days of incubation at 25 °C, in darkness. Experiments of each sample were in triplicate. The holotype specimen was deposited in the Herbarium of the Department of Plant Pathology, Agricultural College, Guizhou University (HGUP). All pure cultures were deposited in the Culture Collection of the Department of Plant Pathology, Agriculture College, Guizhou University, China (GUCC) and in the Mycological Institute of Jilin Agricultural University (HMJAU). All isolates are maintained in 25% (*v*/*v*) glycerol at −80 °C for long-term storage.

### 2.2. Pathogenicity Tests

All isolates were tested for pathogenicity using 3 to 4 cm diameter fruiting body stages of *P. rubrovolvatus* following a modified protocol of Tian et al. [14]; 20 fruiting bodies were sprayed with 0.5 mL spore/cell suspension (1 × 10^6^ spore/cell mL^−1^), while another 20 replicates were sprayed with sterilized distilled water as the controls. These fruiting bodies were maintained under the same conditions (22–24 °C, 90–95% relative humidity) and symptom development was assessed. The pathogenicity test was assessed over 7 days. The strains were re-isolated from the infected fruiting bodies and identified based on morphological and phylogenetic analyses. All experiments were conducted in triplicate.

### 2.3. Morphological, Physiological, and Molecular Characterization

To observe the morphology of the colonies and cells, the pathogens were cultured (1) on a yeast-peptone-glucose (YPG) medium that consisted of 0.5% yeast extract, 1.0% peptone, 2.0% glucose, and 1.6% agar, which were all obtained from Sigma-Aldrich (St. Louis, MO, USA), at 25 °C and (2) in YPG broth at 28 °C in darkness [15,16,17].

Additionally, the physiological and biochemical characteristics of the isolates were determined according to the standard methods described by Kurtzman et al. and Barnett et al. [16,18]. Furthermore, to assess the molecular characteristics of the isolates, total genomic DNA was extracted from the colony of the two pathogenic isolates using a CWBIOTECH Plant Genomic DNA Kit (Beijing, China) following the manufacturer’s protocol. The internal transcribed spacer (ITS) region of the rDNA gene cluster and the D1/D2 domains of the large ribosomal subunit (LSU, 26S) were amplified by PCR with primers ITS4/ITS5 [19] and NL1/NL4 [20].

PCR was conducted in a T100 Thermal Cycler (Bio-Rad Laboratories Inc., Hercules, CA, USA). The total reaction volume of the PCR reaction was 25 µL, consisting of 1.6 µL of dNTP mix (2.5 mM µL^−1^), 0.2 µL of Taq polymerase (5 U µL^−1^), 1 µL of genomic DNA (50 ng µL^−1^), 2 µL of polymerase buffers (10× µL^−1^, Takara, Japan), and 1 µL of each primer (25 mM µL^−1^). Amplification of the ITS region was performed as follows: initial denaturation for 5 min at 94 °C, 30 cycles of 30 s at 94 °C, 30 s at 50 °C, and 30 s at 72 °C, with a final extension for 10 min at 72 °C. Amplification of the fragment of the D1/D2 domains was performed as follows: initial denaturation for 3 min at 94 °C, 36 cycles of 30 s at 94 °C, 30 s at 53 °C, and 60 s at 72 °C, with a final extension for 5 min at 72 °C. Electrophoresis was performed on 0.8% agarose gels stained with Gel Green. PCR products were sequenced by using the same PCR primers used in amplification reactions by Sangon Biotech (Shanghai, China) Co., Ltd.

Kurtzman and Roberts determined the sequence (about 500–600 bp) of the large subunit rRNA gene (26S rDNA) of almost all known yeast taxon. It was found that most species could be distinguished by this marker, and the base difference between different strains within the species was not more than 1% [21,22,23]. The obtained sequences were visualized using BioEdit v7.2.5 [24] and compared to the non-redundant nucleotide collection (nr/nt) sequences present in the National Center for Biotechnology Information (NCBI) GenBank database using nucleotide Basic Local Alignment Search Tool (BLASTn, https://blast.ncbi.nlm.nih.gov/Blast.cgi, accessed on 10 March 2021). Phylogenetic trees were constructed using maximum likelihood (ML), maximum parsimony (MP), and Bayesian inference (BI) via the CIPRES web portal [25]. The phylogenetic analyses were conducted using 40 strains, including our two strains, 18 species of *Saccharomycopsis*, 2 unclassified *Saccharomycopsis* spp., and phylogenetically related species of *Candida*, *Alloascoidea*, *Ascoidea*, and *Wickerhamomyces* [17,23].

### 2.4. Whole-Genome Sequencing and Assembly

Total DNA of the causal agent strain GUCC 202006 (2020060402–1) was extracted using the NuClear Plant Genomic DNA kit (Tiangen Biotech, Beijing, China). We constructed 20-kb libraries, and the genome was sequenced with a PacBio Sequel long-read sequencing platform. RS_HGAP_Assembly.4 protocol was used for assembly and Quiver for genome polishing in SMRT Analysis v3.2.0 [26,27]. High-throughput sequencing on an HiSeq PE150 system (Illumina, San Diego, CA, USA) was carried out to correct the base errors caused by the assembly of long reads from the PacBio SEQUEL using Pilon v1.22 [28]. We assessed integrity at both ends of scaffolds by telomeric repeats (TTAGGG/CCCTAA) [29].

### 2.5. Gene Prediction and Genome Annotation

Augustus v3.2.2 was used for denovo prediction of the protein-coding genes as Stanke et al. [30] (--genemodel=complete --gff3=on --species=haishen haishen.yanzheng.fasta > augustus.gff). Repeat Masker v4.0.5 was used to identify and mask the repeat sequences [31]. Tandem repeat sequences were identified by the Tandem Repeats Finder v4.07b [32]. The predicted protein-coding genes were functionally annotated by BLASTP (blastp -db $swiss -query $faa -out $prefix.swiss.bsp -outfmt 0 -evalue 1e-5 -num_threads 40 -num_descriptions 5 -num_alignments 5 &) query against eleven databases, such as the NCBI Non-redundant (nr) [33], Gene Ontology (GO) [34], Eukaryotic Orthologous Groups (KOG) [35], Pfam [36], Kyoto Encyclopedia of Genes and Genomes (KEGG) [37,38], the SwissProt database [39], Transporter Classification Database (TCDB) [40], Pathogen-Host Interactions Database (PHI-base) [41], Carbohydrate-active enzymes (CAZymes) [42], and Cytochrome P450 monooxygenase (P450) [43].

### 2.6. Orthological, Phylogenetic Tree Construction

OrthoMCL software (v.2.0.9) [44] and all-versus-all BLASTP [45] were used with the parameters (E-value ≤ 1 × 10^−15^, coverage ≥ 50%) to identify the orthologous gene families based on the coding sequence (CDSs) of *S. phalluae*, *S. fermentans*, *S. schoenii*, *S. fodiens*, *S.* sp. UWO(PS) 91-127.1, *S. fibuligera*, *S. malanga*, *S. capsularis* and 20 other selected genomes were obtained from the NCBI database (Appendix A). The shared single-copy genes were screened and aligned using Clustal omega [46]. The proTest was used to generate an optimal base substitution model with maximum likelihood (ML) algorithm in RA × ML [47], with *Scheffersomyces stipitis* CBS 6054 as the outgroup.

## 3. Results

### 3.1. Pathogen Isolation, Pathogenicity Tests and Identification

Yellow rot disease is a severe and extensive disease of *P.*
*rubrovolvatus*, infecting up to 60% of the fruiting bodies (Figure 1B–D). At the early stage, the disease is characterized by reddish water droplets on the surface of the fruiting bodies. Initially, a lot of droplets appear on the surface of the fruiting bodies, and the whole fruiting body then decays. Additionally, many other microorganisms grow with the development of the disease, such as bacteria, fungi, myxomycetes, nematodes, insects, and mites. The disease can spread rapidly to adjacent fruiting bodies resulting in abnormal *P.*
*rubrovolvatus* growth and harvest failure.

Among the 28 isolates that were obtained from the diseased fruiting bodies, only strains GUCC 202006 (2020060402-1) and GUCC 202007 (2020060503-2) were pathogenic (Appendix A). The pathogenicity results showed that yellow rot disease symptoms were visible 3 days after inoculation with the spore suspension, with clear symptoms developing at 4 days post-inoculation. The whole fruiting body decayed at 7 days after inoculation. These symptoms from artificial inoculation were similar to those observed in the field (Figure 1E–G). The control were without disease on the 7 day (Figure 1H). To fulfill Koch’s postulates, the pathogens were consistently re-isolated from the infected fruiting bodies of *P. rubrovolvatus* and confirmed to be consistent with the inoculated strain based on morphological and molecular characteristics.

The physiological and biochemical characteristics were identical between the two strains. The key characteristics of the proposed novel species and related type strains in the genus *Saccharomycopsis* are compared in Table 1.

The sequences of the two isolates, GUCC 202006 and GUCC 202007, were identical. The ITS and LSU sequence data were deposited in GenBank (GUCC 202006 (2020060402-1), type strain: ITS MW412929 and LSU MW405828; GUCC 202007 (2020060503-2): ITS MW412927 and LSU MW405829).

The phylogenetic analyses based on LSU sequences showed that our two strains clustered together with high statistical support (ML/MP/BI: 100%/99%/1.00). The novel species is the sister of *Saccharomycopsis oxydans* IBRC M-30374^T^, based on high statistical support (92%/100%/1.00) (Figure 2). Based on the morphology, physiology and phylogeny results, the two pathogenic strains are introduced as *Saccharomycopsis*
*phalluae* sp. nov.

***Saccharomycopsis phalluae* FH Tian, XX Yuan & KQ Peng, sp. nov**.

**Diagnosis****:** The cells infected fruiting body of *Phallus rubrovolvat**us*, causing yellow rot disease. Characterized with multi-polar budding and septate hyphae. Cells were ellipsoidal to elongate. The margins on cultures were regular. Additionally, the colony could turn reddish-brown in the center after 5 days on YPG agar.

**Holotype:** Voucher number-HGUP 20064 (deposited in the Herbarium of the Department of Plant Pathology, Agricultural College, Guizhou University, China), collected by F.H. Tian, X.X. Yuan and K.Q. Peng on 10 March 2020 at Nayong county, Guizhou Province, diseased fruiting body from a mushroom cultivation base (Figure 1B,C).

**Isotype:** GUCC 202006, GUCC 202007, deposited in GUCC, China. ITS region sequence = GenBank accession no. MW412929, MW412927, respectively; LSU sequence = GenBank accession no. MW405828, MW405829, respectively.

**Type locality:** Nayong county, Guizhou Province, China (105°20′31″ N, 26°53′41″ E).

**Etymology:** Named after the host genus from which it was collected, *Phallus*.

**Chinese name:** Zhu Sun Sheng Kou Nang Fu Mo Jiao Mu 竹荪扣囊复膜酵母

**Distribution:** China, Guizhou Province, Bijie City, Nayong county.

**Habitat:** Cells on the diseased fruiting body of *Phallus rubrovolvatus*.

**Indexfungorum**: IF558187

Morphological description of the pathogen after growth on YPG agar for 3 days at 25 °C: streak culture is white, smooth, glossy, butyrous, convex. Colony margins are regular (Figure 3A,B). The colony turned reddish-brown in the center after 5 days (Figure 3B,C). The cells are ellipsoid to elongate, and measure 5.6–10.4 × 2.4–4.8 μm (av. 7.6 μm × 3.7 μm, *n* = 25), and occur singly or occasionally in pairs. After 15 days, pseudohyphae can be observed at the margin of the colony, and they have a consistent form after subculturing (Figure 3C,D,H–L).

Morphological description of the pathogen after growth in YPG broth for 2 days at 28 °C in darkness: cells are ellipsoid to elongate and occur singly or in pairs (Figure 3E–G). No sexual structures observed. Regarding fermentation ability: negative for D-glucose, maltose, sucrose, lactose, and raffinose. Potassium nitrate, sodium nitrite, glucosamine, cadaverine, and imidazole were assimilated. Growth is weakly positive at 10 °C but positive at 20, 24, 26, 28, 30, and 32 °C after 3 days. No growth is detected at 35 or 37 °C. Growth is positive in the presence of 0.1% cycloheximide. No growth occurs on 5% or 10% NaCl. Starch-like compounds are not produced.

Molecular characteristics (type strain, 2020060402-1^T^): nucleotide sequences of the ITS region (accession no. MW412929) and the D1/D2 domains (accession no. MW405828) of the LSU (26S) were deposited in NCBI GenBank.

**Type:** China, Guizhou Province, isolated from diseased *Phallus rubrovolvatus* (edible and medicinal fungi, Basidiomycete), Fenghua Tian [GUCC 202006] (holotype GUCC 202006; ex-type culture HMJAU 201004-20).

**Note:** Phylogenetic analysis based on LSU sequence data showed that our strains were the sister species of *S**. oxydans* S. Nasr and A. Yurkov, but with 1.8% (9/505 nucleotides) variation. Morphologically, our strains differ from *S. oxydans* by having ellipsoidal to elongate cells instead of ovoid to short cylindroid cells. The margins on cultures of *S. oxydans* are irregular while margins on cultures of *S.*
*phalluae* are regular. Additionally, the colony of our strains turned reddish-brown in the center after 5 days on YPG agar, while *S. oxydans* lacks this characteristic. In the phylogenetic analyses, 18 species of *Saccharomycopsis* were included, as the LSU sequence data of *S. phaeospora* was not available. However, *S. phaeospora* has cells with truncate base rather than broad base [48].

### 3.2. Features of the S. phalluae Genome

In total, 96,001 reads number (600× Depth) were obtained, from which a 14.148 Mb assembly was estimated. The genome consisted of 8 contigs with N50 of 1,822,654 bp, N90 of 1,540,684 bp, and 43.55% G+C content (Table 2), among which contig1, 3, 4 and 5 include complete 5’ and 3’ terminal telomere structure, and contig 2, 6, 7 and 8 only have 3’ terminal telomere structure.

A BLAST search of repeat sequences yielded 12,316,384 bp covering 87.05% of the *S. phalluae* genome; meanwhile, short interspersed nuclear elements (SINE) accounted for 0.06% of the genome. Approximately 3.65% of the genome was long terminal repeats (LTRs), 0.33% was DNA transposons, and 3.91% was long interspersed nuclear elements (LINEs), while minisatellite and microsatellite DNA accounted for 0.07% of the genome.

### 3.3. Functional Annotation of S. phalluae

There were 5966 gene models predicted in the different databases, accounting for 46.56% of the whole genome with an average sequence length of 1,480 bp (Appendix A). We predicted 232 tRNAs (18,914 bp), 2 rRNAs (240 bp), 12 snRNAs (1015 bp), 2 miRNAs (204 bp), and 8 others (1706 bp).

Annotation was performed with the NCBI nr, KEGG, GO, KOG, TCDB, PHI-base, CAZy, and P450 databases (Table 3 and Appendix A). There were 4,735 Non-redundant proteins found in *S. phalluae*. They matched closest with *Ascoidea rubescens* (1252), *Wickerhamomyces anomalus* (433), *Pachysolen tannophilus* (311), *Wickerhamomyces ciferrii* (286), *Cyberlindnera fabianii* (251), *Babjeviella inositovora* (180), and *Kuraishia capsulata* (160), which accounted for 60.68% of total nr predicted genes (Figure 4).

There were 4015 proteins assigned to NCBI KOG categories (Figure 5). The “General function prediction only” category had the most enriched genes (572), followed by “Posttranslational modification, protein turnover, chaperones” (428), “Translation, ribosomal structure and biogenesis” (290), “Intracellular trafficking, secretion, and vesicular transport” (276), and “Signal transduction mechanisms” (259). The representation of genes related to protein and energy metabolism may reflect the capacity of *S. phalluae* to absorb and transform nutrients from a variety of substrates.

Predicted genes were mapped to the KEGG database and assigned functional classifications to 1968 gene models (32.99% of the total gene models, 5966) (Figure 6). Some categories related to protein families and metabolism were highly enriched including “Genetic information processing” (640), “Metabolism” (285), “Signaling and cellular processes” (165), “Carbohydrate metabolism” (119), “Amino acid metabolism” (96), “Metabolism of cofactors and vitamins” (79), “Lipid metabolism” (65) and so on. There also found 155 Unclassified: metabolism.

The TCDB database was to perform protein domain analysis and assigned 290 putative transport proteins to 7 functional classes including “Accessory factors involved in transport”, “Channels/pores”, “Electrochemical potential-driven transporters”, “Group translocators”, “Incompletely characterized transport systems”, “Primary active transporters”, and “Transmembrane electron carriers” (Figure 7). The top two enriched categories were “P-P bond hydrolysis-driven transporters” (290) and “Porters (uniporters, symporters, antiporters)” (271).

In terms of GO functional classes, 3921 proteins were predicted accounted for 65.72% of the total predicted genes in *S. phalluae*. The top eight most highly enriched proteins were predicted in terms of GO functional classes, that accounted GO terms were “Binding”, “Catalytic activity”, “Metabolic process”, “Cellular process”, “Single-organism process”, “Cell”, “Call part”, and “Biological regulation” (Figure 8).

Amino acid sequences were mapped with PHI-base and identified 1779 candidate pathogenicity-related proteins (Figure 9). The “Reduced virulence” category had the most enriched proteins (799), followed by “Unaffected pathogenicity” (402), “Effector (plant avirulence determinant)” (210), together these represented 79.31% of all proteins predicted with PHI-base.

Cytochrome P450 (CYP) is a superfamily of hemoproteins that use heme as a cofactor. CYPs have various substrates in different enzymatic reactions and are present in all kingdoms. Eighty-three putative CYPs genes were identified in *S. phalluae* through a BLAST search that was classified into 24 families. The CYP51 family had the highest number of enriched genes (100), followed by CYP715 (42), and CYP53 (34).

Pathogens can primarily use CAZymes to destroy the polysaccharide component of the host cell wall during the beginning of infection [49]. As *Saccharomycopsis phalluae*, this study confirmed that it was the pathogens of yellow rot disease on *P. rubrovolvatus*. We searched the CAZy database for CAZymes, carbohydrate-binding modules, and auxiliary proteins in the 2020060402-1 genome. A total of 220 CAZyme-encoding gene models were assigned across the six categories of CAZymes, including Glycosyl transferases (GTs; 92), Glycoside hydrolases (GHs; 85), Auxiliary activities (AAs; 21), Carbohydrate esterases (CEs; 15), Carbohydrate-binding module (CBMs; 4) and Polysaccharide lyases (PLs; 3) (Table 4). Based on the study of Xu et al. [50], most genes encoded GH and GT enzymes, might be used to degrade the host cell barrier during the fungi–fungi infection process, across three mushroom pathogens (*Cladobotryum dendroides*, *C. protrusum*, and *Mycogone perniciosa*). Significantly, there were 177 gene models predicted in *S. phalluae* genome, accounting for 80.45% of the total.

### 3.4. Phylogenomics Analysis of S. phalluae

There were 1584 orthogroups with all species present. We next analyzed 794 single copy orthogroups that were conserved across all of the fungi analyzed (Figure 10). The phylogenetic analysis indicated that our new collection, *S. phalluae*, clustered with *S. fodiens*, whereas *Saccharomycopsis* (Saccharomycopsidaceae) is distantly related to *Ascoidea* (Alloascoideaceae) (Figure 10). The analysis results coincide with those based on LSU rRNA sequences analysis (Figure 2). Unfortunately, the genome sequences of several species in *Saccharomycopsis* have not been obtained.

## 4. Discussion

Yellow rot disease is a disastrous disease of *P. rubrovolvatus*. The diseased tissues rapidly become colonized by massive amounts of contaminating microorganisms so identifying the causal agent has been challenging [5,6,7,8]. To determine the cause of the disease, diseased samples at various stages were collected from the cultivation areas. All isolates were purified and inoculated on healthy fruiting bodies by non-invasive inoculation. The result showed that only the yeast-like fungi were pathogenic. Typical yellow rot disease symptoms were observed 3–7 days after inoculation, whereas the controls remained healthy. Thus, the yeast-like fungi were identified as the cause of yellow rot disease. A phylogenetic tree, inferred by the ML, MP, and BI approach based on the LSU gene sequences, confirmed the two isolates as a new taxon. On the basis of the phylogenetic analysis and morphological characteristics, the causal agent was introduced herein as *Saccharomycopsis*
*phalluae* sp. nov. It is characterized by having ellipsoidal to elongate cells, the colony turning reddish-brown in the center after 5 days on YPG agar and with regular margins on cultures instead of ovoid to short cylindroid cells, without turning reddish-brown and with irregular margins to the closed species *S. oxydans*, respectively. In the phylogenetic tree, there are several unidentified *Candida* spp. clustered with *Saccharomycopsis* spp., indicating their misidentification. However, their taxonomic placements need to be further studied.

It is the first pathogen in Saccharomycopsidaceae that were reported on mushroom in China. However, with the rapid development of the planting industry of *P. rubrovolvatus*, the occurrence of yellow rot disease is becoming more and more serious. The severe disease (yellow rot disease) on *P. rubrovolvatus* occurs extensively in China, and causes major economic losses and hinders in the industry. The occurrence of yellow rot disease on various varieties of *P. rubrovolvatus* in different areas needs further investigation, so as to provide a scientific theoretical basis for clarifying the occurrence and epidemic conditions related to the disease. In the future, disease resistance breeding may be useful to control the disease.

However, in plants and other mushroom cultivation, there is no disease caused by *Saccharomycopsis*. In order to comprehensively analyze the relationship between *S. phalluae* and related species and genera, 27 fungal species were used in the phylogenetic analysis. The result showed that *S. phalluae* formed a distinct branch to *S. fodiens* in the clade of *Saccharomycopsis* genus. This result is consistent with the analysis results based on the LSU sequence. To some extent, this also supports the statement that the LSU gene is used as the classification basis of this group.

A total of 5966 genes were predicted from *S. phalluae* in different databases with genome size of 14.148 Mb, While the genome size of other species in the genus ranges from 12.192 Mb to 19.567 Mb, and the number of annotated genes ranges from 5359 to 6736, respectively (Appendix A). This provides basic data for the analysis of pathogenic genes, biosynthesis and other functional genes of this species. There were 4015 proteins in *S. phalluae* assigned to NCBI KOG categories. The representation of genes related to protein and energy metabolism could reflect the capacity of *S. phalluae* to absorb and transform nutrients from a variety of substrates. It is understood that CAZymes play a relevant role in virulence. Most chitinase- and cellulose-degrading enzymes are categorized within the GH class and the abundance of GH18 was consistent with the efficient degradation of chitinase, cellulose, and hemicellulose [51]. There were 85 GHs and 92 GTs predicted in the genome of *S. phalluae* accounting for 80.45% of the total CAZymes, which may be the pathogenicity related genes that might be used to degrade the host cell barrier (chitinase, cellulose, and hemicellulose, or other organizational structure) during the fungi–fungi infection process [49,50,51]. Of the GH families, GH18 was encoded by the most genes (12) in *S. phalluae*, which suggested that these enzymes might play a role in the genome of our pathogen. There were 21 AA (Auxiliary activities) genes in the *S. phalluae* genome, among them, 9 AA3 (glucose-methanol-choline oxidoreductase) was the most abundant. 

Pathogenic fungi can cause huge damage to the host. The effectors are important virulence determinants of pathogenic fungi and play important role in successful pathogenesis, predominantly through targeting and regulating the phytohormone signaling of hosts by changing or operating them [52]. There are 210 effectors found in *S. phalluae*, which were also found in the pathogens, such as *Fusarium graminearum*, *Magnaporthe oryzae*, *Ustilago maydis*, *Botrytis cinerea*, *Verticillium dahliae*, *Aspergillus fumigatus*, *Cochliobolus carbonum*, *Erwinia amylovora*, *Septoria lycopersici*, *Colletotrichum lindemuthianum*, *Bipolaris sorokiniana* and *Paenibacillus larvae*. Pathogens may optimize their own effector sets to adapt to hosts, as several effector proteins in mushroom pathogens (*C. protrusum* and *M. perniciosa*) [53,54]. In addition, Yin et al. reported that pathogens could secrete many proteins that can support the colonization of the host surface during infection [55]. From the results of analysis on *S. phalluae*, most of the 1779 candidate pathogenicity-related proteins are with PHI-base. These findings could be instrumental for the understanding of fungi–fungi interactions, and for exploring efficient management strategies to control the disease. 

## Figures and Tables

**Figure 1 jof-07-00707-f001:**
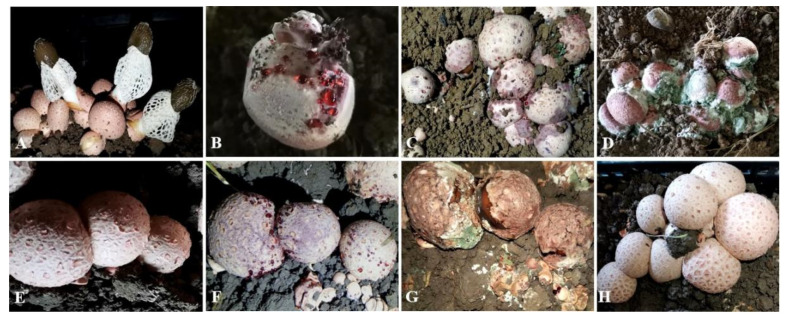
Field symptoms of yellow rot disease on *Phallus*
*rubrovolvatus* and pathogenicity tests of isolate GUCC 202006 (2020060402-1). (**A**) Healthy fruiting bodies of *P.*
*rubrovolvatus*. (**B**–**D**) Field symptoms of yellow rot disease on *P.*
*rubrovolvatus*. (**E**–**G**) Pathogenicity tests, after spraying with 0.5 mL spore suspension (1 × 10^6^ conidia mL^−1^). (**E**) Day 0 after inoculation; (**F**) Reddish water droplets on the surface of the fruiting body, 4 days after inoculation; (**G**) Whole fruiting body rotten, 7 days after inoculation. (**H**) Control, no disease, 7 days after inoculation with distilled water.

**Figure 2 jof-07-00707-f002:**
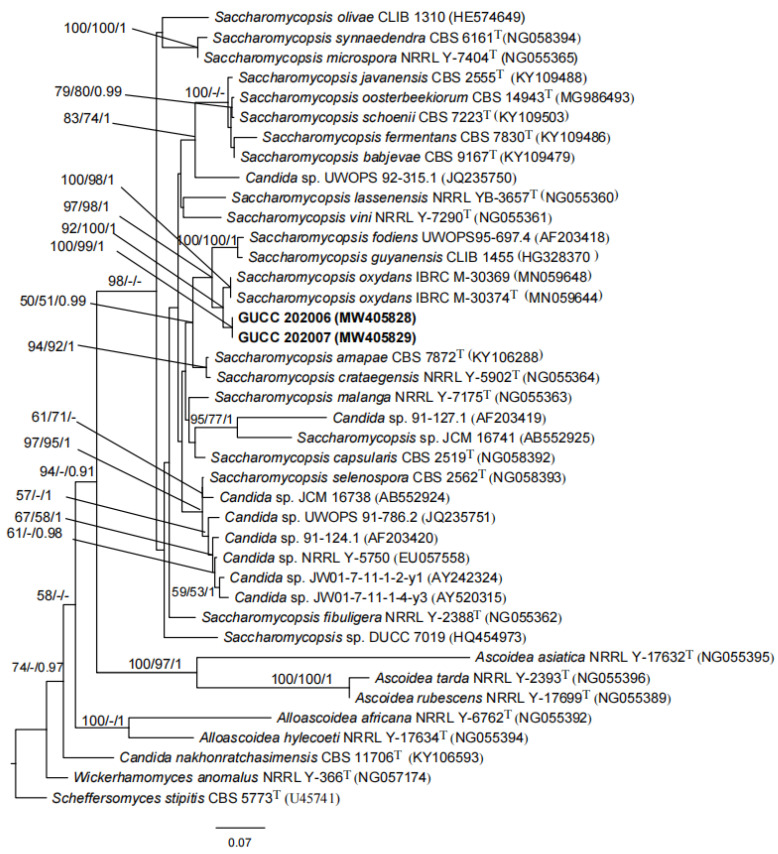
Phylogenetic tree based on LSU gene sequences for our two strains and selected reference isolates retrieved from GenBank. Maximum parsimony (MP) values, maximum likelihood (ML) values > 50% and Bayesian inference (BI) values > 0.90 are shown next to topological nodes and separated by “/”. Bootstrap values < 50% and BI values < 0.90 are labeled with “-”. Ex-type strains are marked with “^T^”. The tree was rooted to *Scheffersomyces stipitis* CBS 5773^T^. The two strains in bold were *Saccharomycopsis phalluae* sp. nov.

**Figure 3 jof-07-00707-f003:**
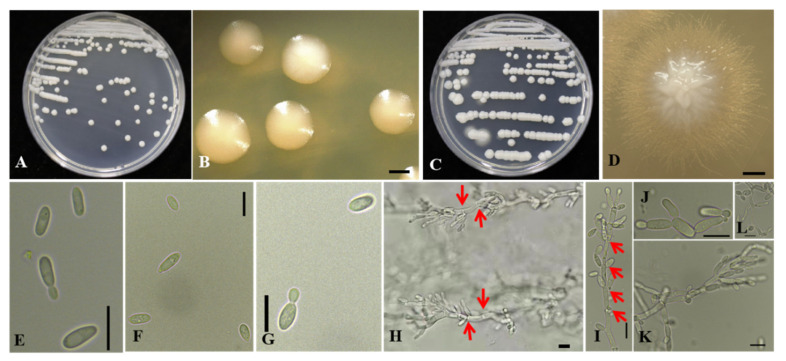
Morphological characterization of *Saccharomycopsis phalluae* sp. nov. GUCC 202006 on YPG agar. (**A**) White colony on YPG agar at 25 °C after 3 days of incubation. (**B**) Regular colony margins, bar = 500 μm. (**C**) Pseudohyphae at the edge of the colony. (**D**) Pseudohyphae exhibiting a consistent form after subculturing, bar = 500 μm. (**E**) Budding cells in YPG broth, bar = 10 μm. (**F**,**G**) Budding cells on YPG broth, bar = 10 μm. (**H**–**L**) Pseudohyphae with septate cells (red arrows: cell membrane) on YPG agar, bar = 10 μm.

**Figure 4 jof-07-00707-f004:**
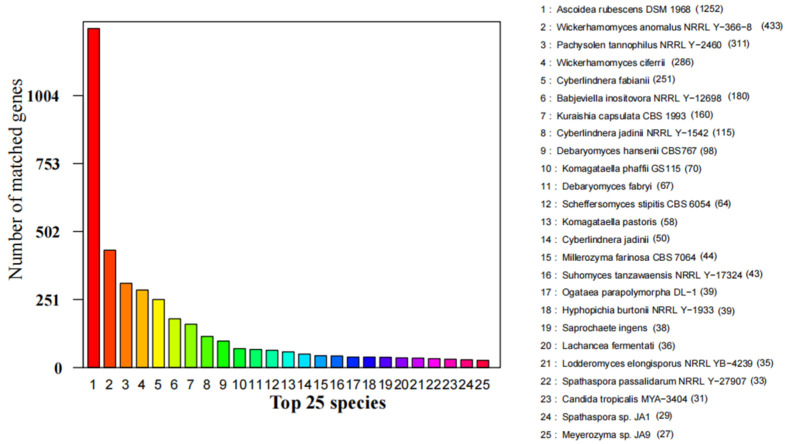
Predicted proteins from *S. phalluae* genome to the NCBI non-redundant proteins database among different fungal species.

**Figure 5 jof-07-00707-f005:**
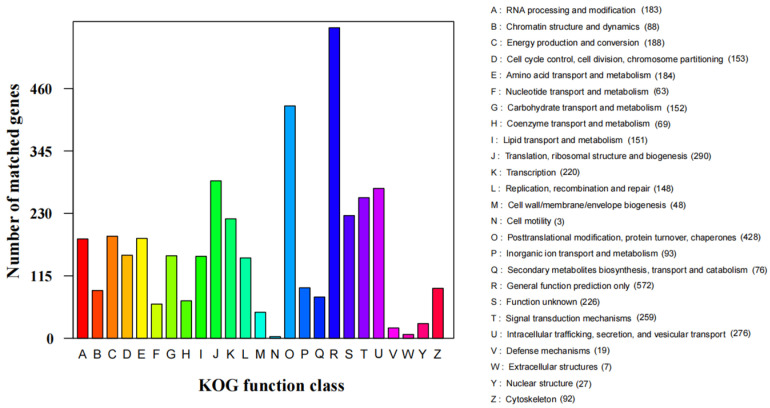
KOG functional classification of *S. phalluae* proteins.

**Figure 6 jof-07-00707-f006:**
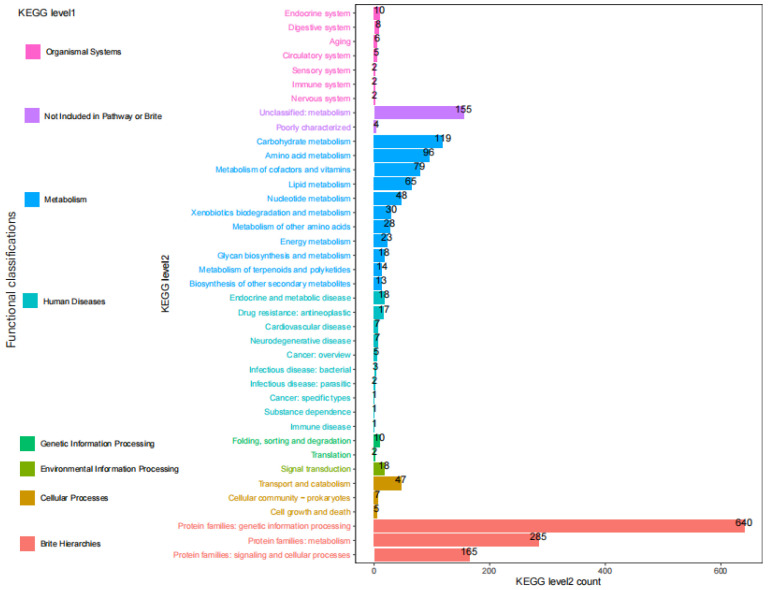
KEGG pathway annotation of *S. phalluae* genes.

**Figure 7 jof-07-00707-f007:**
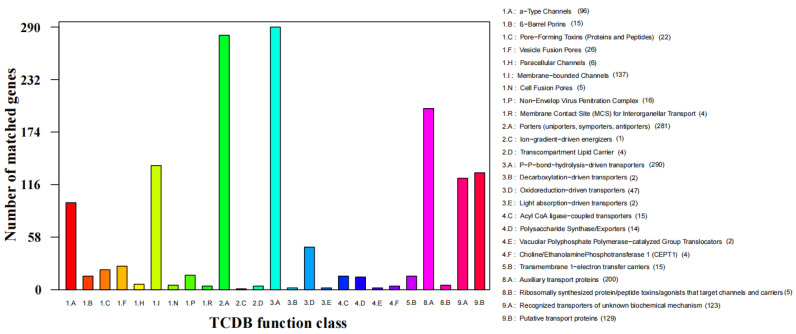
TCDB functional annotation of *S. phalluae* genes.

**Figure 8 jof-07-00707-f008:**
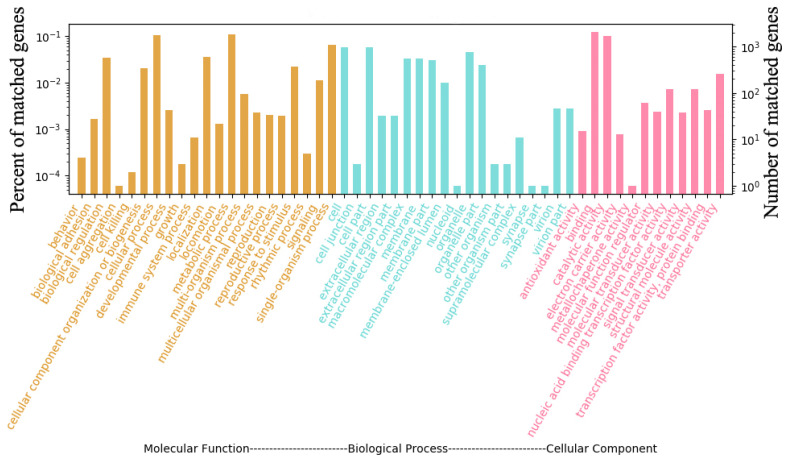
GO functional annotation of *S. phalluae* genes.

**Figure 9 jof-07-00707-f009:**
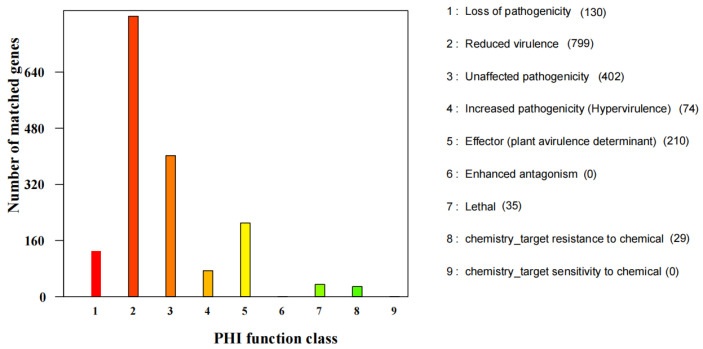
PHI functional annotation of *S. phalluae* genes.

**Figure 10 jof-07-00707-f010:**
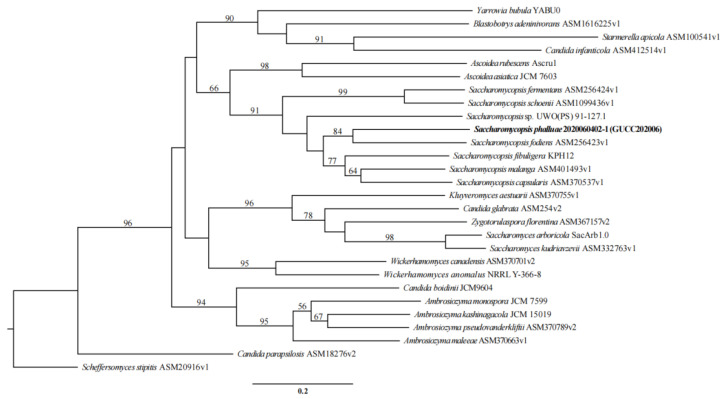
Phylogenetic tree of *S. phalluae* and 27 other fungal species. Maximum likelihood (ML) values > 50% were placed close to topological nodes, with 794 single copy orthologs. The tree was rooted to *Scheffersomyces stipitis* CBS 6054, *Saccharomycopsis phalluae* sp. nov. was in bold.

**Table 1 jof-07-00707-t001:** Physiological characteristics of strain 2020060402-1^T^ and related type strains of *Saccharomycopsis* species.

Characteristic	1	2 ^a^	3 ^b^	4 ^c^	5 ^d^
D-glucose fermentation	−	W/−	W/V	−	−
Maximum growth temperature	35 °C	ND	33 °C	35 °C	28 °C
Growth in 5% NaCl	−	ND	−	ND	+
Growth in 10% NaCl	−	−	ND	+	−
Potassium nitrate	+	−	−	+	−

Taxa: (1) 2020060402-1^T^; (2) *S. crataegensis* strain NRRL Y-5902^T^; (3) *S. fodiens* strain CBS 8332^T^; (4) *S. guyanensis* strain CBS 12914^T^; (5) *S. oxydans* strain IBRC-M 30374^T^. Codes: +, positive; w, weakly positive; −, negative; V, variable; ND, not determined. Data are from this study, except those that are labeled as follows: ^a^ data from Kurtzman et al., (2011); ^b^ data from Lachance et al. (2012); ^c^ data from Jacques et al. (2014); ^d^ data from Hajihosseinali et al. (2020).

**Table 2 jof-07-00707-t002:** General features of the *S. phalluae* genome.

Scaffold Characteristic	
Total counts of contigs sequences	8
Total length of contigs sequences (bp)	14,148,124
Contigs N50 (bp)	1,822,654
Contigs N90 (bp)	1,540,684
Largest scaffold length (bp)	2,335,166
Genome coverage	600×
GC content (%)	43.55
**Genome Characteristic**	
Gene number	5966
Average gene length (bp)	1480
Gene density (Unit:gene_number/1000 bp)	0.4
Exon number per Gene	1.14
Exon average length (bp)	1298
Genome GC percent (%)	43.55
Exon GC percent (%)	46.56

**Table 3 jof-07-00707-t003:** Summary of *S. phalluae* gene annotations.

Database Used for Gene/Protein Annotation	Number of Genes
nr	4735
GO	3921
KEGG	1968
KOG	4015
TCDB	1461
PHI	1779
CAZy	220
P450	264

nr, National Center for Biotechnology Information Non-redundant Protein Database; GO, Gene Ontology; KEGG, Kyoto Encyclopedia of Genes and Genomes; KOG, Eukaryotic Orthologous Groups; TCBD, Transporter Classification Database; PHI, Pathogen–Host Interactions Database; CAZy, Carbohydrate-active Enzymes Database; P450, cytochrome P450 monooxygenase.

**Table 4 jof-07-00707-t004:** Carbohydrate-active enzyme annotation results in *S. phalluae*.

Classification	Number
Carbohydrate-binding module (CBMs)	4
Carbohydrate esterase (CEs)	15
Glycoside hydrolase (GHs)	85
Glycosyl transferase (GTs)	92
Polysaccharide lyase (PLs)	3
Auxiliary activities (AAs)	21

## Data Availability

The genome sequence data and assembly reported in this paper are associated with NCBI BioProject: PRJNA721835, BioSample: SAMN18740300 and Accession Number: CP073212-CP073219 in GenBank.

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
