# Peer review of "Complete Genomic Characterization and Identification of Saccharomycopsisphalluae sp. nov., a Novel Pathogen Causes Yellow Rot Disease on Phallus rubrovolvatus"

_jof, 2021, doi:10.3390/jof7090707_

Round 1

Reviewer 1 Report

The authors have improved the manuscript, however one important piece still needs to be completed. There are 8 species that do not have DNA sequence data so the authors must have at least one morphological distinction separating this new species with those 8 species. Without this, there is no way of knowing that this species is not one of those 8 species without DNA sequences. Review the taxonomical descriptions of those 8 species without sequence data and see if these is at least one difference. It could be that none of those 8 species grow on fungal fruit bodies or that none of those 8 are pathogens, or better yet the cells sizes, colony formation, or growth temperatures are different but this must be done.

Author Response

We would like to thank the editor for giving us a chance, and also thank the reviewers for giving us constructive suggestions which would help us both in English and in depth to improve the quality of the paper. Here we submit a revised of our manuscript with the title “Complete Genomic Characterization and Identification of Saccharomycopsis phalluae sp. nov., a Novel Pathogen Causing Yellow Rot Disease on Phallus rubrovolvatus”, which has been modified according to the reviewers’ suggestions. Efforts were also made to correct the mistakes and improve the English of the manuscript. We mark all the changes in Red in the revised manuscript as a supplementary file.

The following is a point-to-point response to the reviewers’ comments.

Response to Reviewer 1 Comments:

  1. Lower case b =  basidiomycete

Answer: It was modified, Line 35. Please check it. Thank you.

  1. remove commaLine 45.

Answer: It was modified, Line 47. Please check it. Thank you.

  1. consider removing commaLine48

Answer: It was modified, Line 50. Please check it. Thank you.

  1. consider revising to: has not been reported as a plant or mushroom pathogen.

Answer: It was modified, Line 61. Please check it. Thank you.

  1. consider revising: Each basidiocarp was first externally washed with running tap water.

Answer: It was modified, Line 75. Please check it. Thank you.

  1. Consider revising to: Next,

Answer: It was modified, Line 75. Please check it. Thank you.

  1. consider revising to: was

Answer: It was modified, Line 83. Please check it. Thank you.

  1. “the”Delete

Answer: It was modified, Line 84. Please check it. Thank you.

  1. consider revising to: and in the

Answer: It was modified, Line 86. Please check it. Thank you.

  1. consider revising to:(HMJAU). All isolates are maintained

Answer: It was modified, Line 87. Please check it. Thank you.

  1. Question: Some of the isolates have spores but some are yeast. It would be more appropriate to state something like:5 mL spore/cell suspension (1x10 spore/cell mL)

Answer: It was modified, Line 92. Please check it. Thank you.

  1. There are 24 in index Fungorum according to the introduction. Were the other 8 species evaluated against the 2 strains to find morphological differences? This must be done since there is no genetic evidence.

Answer: It was my mistake. At that time, I included several species whose taxonomic status changed to other genera Actually there are 19 species of Saccharomycopsis aceppt in Index Fungorum as of August 25, 2021 (http://www.indexfungorum.org/Names/Names.asp). The phylogenetic analyses were conducted using 40 strains, including our two strains, 18 species of Saccharomycopsis, 2 unclassified Saccharomycopsis spp., and phylogenetically related species of Candida, Alloascoidea, Ascoidea, and Wickerhamomyces. Of the 19 accepted species in Saccharomycopsis genus, S. phaeospora was not included in the phylogenetic analyses as its LSU sequence is no available and S. phaeospora has cells with truncate base while our strains with broad base. The morphological characteristic of our strains are distinct from S. phaeospora. The lack of LSU sequence information of S. phaeospora does not affect the identification results. It was modified, Line 56, Line 133-134, Line287-290. Please check it. Thank you.

  1. consider deleting this section as it is redundant.

Answer: It was modified, Line 185. Please check it. Thank you.

  1. consider deleting the comma

Answer: It was modified, Line 186. Please check it. Thank you.

  1. consider revising to: harvest failure.

Answer: It was modified, Line 187. Please check it. Thank you.

  1. consider deleting this word as it is not needed.

Answer: It was modified, Line 189. Please check it. Thank you.

  1. consider deleting this section as it is redundant:....were pathogenic (Figure S1).

Answer: It was modified, Line 189-190. Please check it. Thank you.

  1. consider deleting as it is not needed.

Answer: It was modified, Line 193. Please check it. Thank you.

  1. Please correct this part.

Answer: It was modified, Line 245. Please check it. Thank you.

  1. Remove“the”

Answer: It was modified, Line 261. Please check it. Thank you.

  1. Remove“the”

Answer: It was modified, Line 268. Please check it. Thank you.

  1. Remove“are”

Answer: It was modified, Line 269. Please check it. Thank you.

  1. consider “,”changing to :

Answer: It was modified, Line 269. Please check it. Thank you.

  1. Remove“, the results are ”

Answer: It was modified, Line 269. Please check it. Thank you.

  1. please consider removing the comma and revise to something like the following:32 C after 3 days.

Answer: It was modified, Line 272. Please check it. Thank you.

  1. Remove“are”

Answer: It was modified, Line 275. Please check it. Thank you.

  1. Consider revising as this is hard to read as written.

Answer: It was modified, Line 302-304. Please check it. Thank you.

  1. consider revising to:They matched closest with...

Answer: It was modified, Line 340. Please check it. Thank you.

  1. consider revising to: accounted

Answer: It was modified, Line 343. Please check it. Thank you.

  1. consider revising to: may

Answer: It was modified, Line 368. Please check it. Thank you.

  1. This is considered informal. Please remove the word we in all the results section to make all these more formal. for example:Predicted genes were mapped to the KEGG databases and assigned...

Answer: It was modified, Line 374, 384, 393, 401. Please check it. Thank you.

  1. This is considered informal. Please remove the word we from the results section:For example: The TCDB database was used to preform protein...

Answer: It was modified, Line 384. Please check it. Thank you.

  1. We needs to be removed throughout the results section:For example: Eighty-three putative CYPs genes were identified in S. phalluae through...

Answer: It was modified, Line 408. Please check it. Thank you.

  1. Remove “totally”

Answer: It was modified, Line 442. Please check it. Thank you.

  1. suggest changing to: consistent

Answer: It was modified, Line 446. Please check it. Thank you.

  1. remove comma

Answer: It was modified, Line 451. Please check it. Thank you.

  1. Maybe it was not clear the first time: The phylogenetic analyses are only on 16 exo-type species. There are still 8 species that do not have sequence data. How do you know that this new species is not one of those 8 species without DNA sequence data. The authors need to look at the morphological data to show that this species has at least one morphological distinct characteristic compared to the other remaining 8. It is likely that this species is distinct since the author's state that pathogen of a mushroom might be considered a new habitat, but the authors do not make any distinction of how to tell this new species from the 8 species that do not have DNA sequence data. The authors must do this.

Answer:  It was modified, Line 56, Line 133-134, Line287-290. Please check it. Thank you.

  1. Please remove as so on is not informative.Consider revising to: Bipolaris sorokiniana, and Paenibacillus larvae.

Answer: It was modified, Line 510. Please check it. Thank you.

Reviewer 2 Report

Overall, the manuscript is well written and structured. The content is relevant to those working to this specific group of yeasts. A new genome assembly is always useful for future studies. I recommend an acceptance after the following minor corrections.

Title: “Identification”

L73: How many samples were collected?

L80: “All experiments were in triplicate” This statement is unclear. Which experiments? Do the authors mean that three samples, each at a different degree of disease, were collected, cultured, and single-spore isolated? Three single-spore isolates were obtained from each original sample? The original samples were all taken from a same specimen of Phallus rubrovolvatus or from three different specimens?

L92: the pathogeniticity test was assessed during how many days?

L122: “this sequence” should be changed to “this marker”.

L139-L140: “the base errors caused by the assembly of long read long sequences in the PacBio Sequel long read sequencing by using Pilon v1.22”. The sentence can be simplified to “the base errors caused by the assembly of long reads from the PacBio SEQUEL using Pilon v1.22”.

L191: “…confirmed to be consistent with the inoculated strain.”…How? isolate and resequence to check whether the exactly same sequence was obtained?

L414: the section 3.4 should be named “Phylogenomics…” instead of “Phylogenetics…”

L424: “coincide”, not “are coincide”.

Author Response

We would like to thank the editor for giving us a chance, and also thank the reviewers for giving us constructive suggestions which would help us both in English and in depth to improve the quality of the paper. Here we submit a revised of our manuscript with the title “Complete Genomic Characterization and Identification of Saccharomycopsis phalluae sp. nov., a Novel Pathogen Causing Yellow Rot Disease on Phallus rubrovolvatus”, which has been modified according to the reviewers’ suggestions. Efforts were also made to correct the mistakes and improve the English of the manuscript. We mark all the changes in Red in the revised manuscript as a supplementary file.

The following is a point-to-point response to the reviewers’ comments.

Response to Reviewer 2 Comments:

Overall, the manuscript is well written and structured. The content is relevant to those working to this specific group of yeasts. A new genome assembly is always useful for future studies. I recommend an acceptance after the following minor corrections.

  1. Title: “Identification”

Answer: It was modified, Line 2. Please check it. Thank you.

  1. L73: How many samples were collected?

Answer: It was modified, Line 72-73. Please check it. Thank you.

  1. L80: “All experiments were in triplicate” This statement is unclear. Which experiments? Do the authors mean that three samples, each at a different degree of disease, were collected, cultured, and single-spore isolated? Three single-spore isolates were obtained from each original sample? The original samples were all taken from a same specimen of Phallus rubrovolvatus or from three different specimens?

Answer: It was modified, Line 82. Please check it. Thank you.

  1. L92: the pathogeniticity test was assessed during how many days?

Answer: It was modified, Line 95-96. Please check it. Thank you.

  1. L122: “this sequence” should be changed to “this marker”.

Answer: It was modified, Line 126. Please check it. Thank you.

  1. L139-L140: “the base errors caused by the assembly of long read long sequences in the PacBio Sequel long read sequencing by using Pilon v1.22”. The sentence can be simplified to “the base errors caused by the assembly of long reads from the PacBio SEQUEL using Pilon v1.22”.

Answer: It was modified, Line 143-144. Please check it. Thank you.

  1. L191: “…confirmed to be consistent with the inoculated strain.”…How? isolate and resequence to check whether the exactly same sequence was obtained?

Answer: It was modified, Line 200-201. Please check it. Thank you.

  1. L414: the section 3.4 should be named “Phylogenomics…” instead of “Phylogenetics…”

Answer: It was modified, Line 436. Please check it. Thank you.

  1. L424: “coincide”, not “are coincide”.

Answer: It was modified, Line 446. Please check it. Thank you.

Round 2

Reviewer 1 Report

The authors have done a good job addressing my major concern. There are only minor changes. Thank you for your efforts.

Line 89: cel should be cell

Line 92: I'm not sure "during" is the correct word. Did the authors mean "after" or "over"? Consider revising if it makes sense.

Line 278: However, S. phaeospora has cells with truncate base rather than broad base[48].

Thank you very much, this is the information that was required, wonderful!

Line 279-280: So, we introduced our......

Please consider removing, this is not needed and is typically not added to species descriptions.

Author Response

We would like to thank the editor for giving us a chance, and also thank the reviewers for giving us constructive suggestions which would help us both in English and in depth to improve the quality of the paper. Here we submit a revised of our manuscript with the title “Complete Genomic Characterization and Identification of Saccharomycopsis phalluae sp. nov., a Novel Pathogen Causing Yellow Rot Disease on Phallus rubrovolvatus”, which has been modified according to the reviewers’ suggestions. Efforts were also made to correct the mistakes and improve the English of the manuscript. We mark all the changes in yellow background in the revised manuscript as a supplementary file.

The following is a point-to-point response to the reviewers’ comments.

Response to Reviewer 1 Comments:

  1. Line 89: cel should be cell

Answer: It was modified, Line 92. Please check it. Thank you.

  1. Line 92: I'm not sure "during" is the correct word. Did the authors mean "after" or "over"? Consider revising if it makes sense.

Answer: It was modified, Line 96. Please check it. Thank you.

  1. Line 278: However, S. phaeospora has cells with truncate base rather than broad base[48].

Thank you very much, this is the information that was required, wonderful!

Answer: Thank you for your guidance.

  1. Line 279-280: So, we introduced our......

Please consider removing, this is not needed and is typically not added to species descriptions.

Answer: It was modified, Line 290. Please check it. Thank you.

This manuscript is a resubmission of an earlier submission. The following is a list of the peer review reports and author responses from that submission.

Round 1

Reviewer 1 Report

The present manuscript describes a study conducted to isolate, identify and sequence the complete genome of Saccharomycopsis phalluae, a new species identified as the causing agent of yellow rot disease on Phallus rubrovolvatus.

My main concerns are related with the relevance of some of the text. There are sentences that would be more appropriate as an introductory text and there is little exploration/description of the results and, more importantly, their critical discussion and comparison with other species that are mushroom pathogens (this last point might imply data re-analysis). My advice is that the text is reviewed and, in particular, the Discussion is re-written.

Furthermore, the description of the new species should be improved: structured in a more formal way and taking into consideration the International Code of Nomenclature for algae, fungi, and plants. 

Some points in the methodology need to be clarified.

The reference list could be improved as some references are not accessible or are not the most relevant for a given sentence.

A more detailed review is presented in the attached file.

Reviewer 2 Report

Complete Genomic Characterization and Identification of Saccharomycopsis phalluae sp. nov., a Novel Pathogen Causing Yellow Rot Disease on Phallus rubrovolvatus.

The manuscript discusses a new species of Saccharomycopsis that causes disease on Phallus rubrovolvatus. The authors demonstrate that this species causes the Yellow Rot, obtained a genome, and describes it as a new species. 

1) The strength of the manuscript is that this disease causes loss of food production and the genomic data would be of value.

2) The weakness of the manuscript is that it is very descriptive of the general outcome of the genome, but does not compare how it relates to other Saccharomycopsis genomes or other species within the Saccharomycotina. This is not the first Saccharomycopsis genome to be published so at least the authors should compare their results to what has been previously published within the genus. (see https://mycocosm.jgi.doe.gov/Sacca1/Sacca1.info.html). This information could be presented within the discussion to make it longer.

For example, was this genome size similar to other Saccharomycopsis or other species with the Saccharomycotina? Are there any genes over represented or under-represented as compared to other Saccharomycopsis species? other genera within the Saccharomycotina? Most of the genomic description would be more informative as a table rather than a long narrative.

3) The results and discussion section are separate; however there are citations within the results (line 176, line 401, line 424) and statements that reflect discussion (lines 353-355, lines 393-394, lines 422-426, some of lines 441-446 discussion or methods section more appropreate).

4) Maybe I missed it but was the genome information deposited? If not the manuscript should not be published until it is deposited in the NCBI database or other public repository?

5) Do all Saccharomycopsis species have DNA sequences in NCBI? If not, the authors need to address species without DNA sequences by discussing how this new species is morphologically different from these Saccharomycopsis because they were not compared in the phylogenetic analyses. This needs to be stated within the manuscript.

Overall: This manuscript has potential but the authors need to deposit the genome information (unless I missed it), need to make sure the morphology of this new species is different from any Saccharomycopsis species without DNA and add to the manuscript, clean the results to include only results, review other yeast species descriptions and make this one similar (suggest looking over some in Fungal planet descriptions for style), and increase the discussion by comparing the genomic results of obtained in their study to other genomes of Saccharomycopsis and Saccharomycotina. Below are some helpful links.

https://mycocosm.jgi.doe.gov/Sacca1/Sacca1.info.html

https://mycocosm.jgi.doe.gov/clm/run/Sacca1-comparative-qc.2664;pKn7Vd?organism=Sacca1

Other specific line items:

Key words:

Key words should be words that are not contained within the title. Consider removing yellow rot disease and add something else.

Introduction:

Line 38: Sentences should not start with an abbreviation.

Please spell out Phallus and check the rest of the manuscript for similar issues.

Line 61: plant plant and mushroom.

The word plant has been typed twice.

Material and methods:

Line 79-80: What were the lighting conditions during incubation? Please add for repeatability.

Line 97-98: Sentence is all alone as a paragraph; please consider adding it to the previous paragraph.

Line 124: Ascoidea needs to be in italics.

Line 131: was carried out to correct by using...

This sentence seems to be missing something, please review and correct.

Line 135: was used to do denovo prediction...

Please consider revising: was used for denovo prediction...

Line 152: Please define CDSs as this is the first and only usage but most people will not know what CDSs is.

Line158-172: All this is instructions to authors and needs to be removed.

Line 253-266: The type morphology needs to be worded more appropriately and in a concise manner. I suggest reviewing other Saccharomycopsis species descriptions or other yeast descriptions and addjusting.

Line 326:..had most matching with seven...

Please consider revising as it is awkward as written.

Line 349: Reword sentence so as not to start with a number.

Line 359-360: assigned functional classification to 1,968 (xxxxxx).

Please define 1,968?  contigs? genes? clusters?

Discussion: Please add additional information about how the genome relates to other similar studies (genomes of Saccharomycopsis and Saccharomycotina).